# Immunophenotypic and Functional Interindividual Variability in Banked Cord Blood Cells: Insights for Advanced Therapies

**DOI:** 10.3390/ijms26031208

**Published:** 2025-01-30

**Authors:** Diana María Vanegas Lozano, Bellaneth Devia Mejia, Catalina Machuca Acevedo, Valentina Jaramillo Mejia, Andrea Marisol Moreno González, Anita Krisko, Sandra Milena Quijano Gómez, Ana María Perdomo-Arciniegas

**Affiliations:** 1Cord Blood Bank, Instituto Distrital de Ciencia, Biotecnología e Innovación en Salud, Bogotá 111611, Colombia; diana.vanegas21@gmail.com (D.M.V.L.); bdevia@idcbis.org.co (B.D.M.); catalina.machuca88@gmail.com (C.M.A.); vjaramillo@idcbis.org.co (V.J.M.); amoreno@idcbis.org.co (A.M.M.G.); 2Department of Experimental Neurodegeneration, University Medical Center Gottingen, 37037 Gottingen, Germany; kriskoanita@gmail.com; 3Grupo de Inmunobiología y Biología Celular, Departamento de Microbiología, Pontificia Universidad Javeriana, Bogotá 110231, Colombia; squijano@javeriana.edu.co

**Keywords:** hematopoietic stem cells, EuroFlow, hematology analyzer, advanced cellular therapies, migration profiles

## Abstract

Umbilical cord blood (UCB) is an alternative therapeutic resource for treating both hematological and non-hematological diseases, especially for pediatric patients. However, UCB transplantation faces challenges, including delayed engraftment, increased risk of graft failure, and slower immune recovery. To maximize its clinical potential, it is essential to understand the variability and functionality of its nucleated cells. This study focused on characterizing UCB cellular populations, viability, and functionality at three key processing stages: freshly collected, post-volume reduction, and post-thawing. Using EuroFlow-based flow cytometry, significant changes were observed in granulocyte and T-cell populations during processing. Additionally, integrating EuroFlow data with hematology counts revealed variability that could affect the yield of specific cell populations, potentially influencing therapeutic decisions. An in vitro migration assay, designed to mimic the vascular niche, was employed to study donor variability in cellular migratory patterns. Notably, thawed UCB cells displayed two distinct migration profiles, distinguishing lymphocyte-like cells from monocyte-like cells. These findings underscore the importance of reproducible cellular quality control measures, such as immunophenotypic and functional donor characterization, to ensure the integrity of UCB composition. A better understanding of these parameters could improve the consistency and reliability of UCB as a starting material for the development of advanced therapies.

## 1. Introduction

In 1988, Dr. Eliane Gluckman performed the first umbilical cord blood (UCB) transplant from a Human Leukocyte Antigen (HLA)-identical sibling as a treatment for Fanconi anemia [1], and, during the following years, the path for this hematopoietic progenitor and stem cell (HPSC) source started an accelerated process, including the establishment of UCB banks in the United States and Europe between 1992 and 1993 [2]. Although several issues have been addressed throughout these years to improve the overall survival and disease-free survival rates of patients transplanted with UCB [1], delayed engraftment, high risk of graft failure, and delayed immune reconstitution remain the principal clinical problems for these patients compared to other sources, such as bone marrow or mobilized peripheral blood, as a consequence of the lower cell dose of HPSC in UCB grafts [3]. The almost simultaneous emergence of haploidentical transplant, which followed a similar process of improving pitfalls until reaching the T-cell-depleted graft to overcome the higher incidence of graft versus host disease, has acquired greater importance during the last decade, even surpassing the number of UCB transplants performed per year, at least in the United States and Europe [1,4,5]. However, the clinical comparison between UCB and haploidentical graft sources probably requires further development, with prospective clinical trial designs, equality of HLA matching between the graft and the donor, and more accurate UCB selection to properly evaluate the outcomes with both sources [6]. Some authors have pointed out that the replacement of UCB allogeneic donor use for a haploidentical donor in hematopoietic transplant could also be related to logistic and economic factors rather than clinical outcome differences [7].

According to the World Marrow Donor Association (WMDA), more than 800,000 UCB units from public banks are currently available worldwide for transplantation, as reported by donor registries [8]. Although UCB graft use has decreased during the last few years, the expertise and cell resources in cord blood banks acquired these past decades might be highly valuable in two different scenarios: either to optimize the clinical results obtained with HPSC transplants nowadays through a personalized approach or as a source for other types of cell therapy in regenerative medicine [7,8,9]. Another potential research field is found in the remaining units not available for transplant due to cell number count, either fresh or frozen, which might be substantial in all UCB banks. Mantri et al. recently proposed a model to use the otherwise discarded fresh units by stakeholders, such as research groups in universities, whose study topics benefit the stem cell and regenerative medicine fields [10].

The Public Umbilical Cord Blood Bank in Bogotá, Colombia started operations in 2014 [11] and currently has more than 1272 UCB units in the inventory available for transplant and over 3000 frozen units that do not fulfill the quality requirements for clinical purposes. As bank research and development activities continue, and following the pathways proposed by the aforementioned authors regarding the potential use of UCB units in other fields, we want to explore the possibilities of using our inventory as an important cell source for advanced therapy development and implementation. However, a common limitation of these therapies is the variability that is inherent to the donor and that might impact differentially the cellular populations present in the UCB units.

Therefore, the cell populations in different stages of the processing of UCB units—collection, volume reduction, cryopreservation, and thawing—were compared in terms of phenotype characterization, viability, and functionality. Characterization was conducted using EuroFlow protocol analysis, which is a standardized flow cytometry test for diagnostic and prognosis purposes, and a classical analysis using an automatized hematology analyzer. Viability and functionality assessments were performed using a canonical niche stimuli migration assay, which aims to mimic the intramedullary migration. The Euroflow protocol analysis showed more consistent and informative data in terms of qualitative and quantitative reports of cellular populations in the UCB units throughout the banking processes compared to the classical measurement using an automatized hematology analyzer. The variability in the cell population proportions and the total cellular counts increases after thawing, especially for granulocytes and lymphocytes, as well as their viability and functionality. We identified using migration assays different profiles of functionality. Our results show that it is helpful to address the variability in UCB units due to processing to improve hematopoietic transplantation outcomes or to develop guidelines of UCB cell populations used in off-the-shelf advanced cellular therapies.

## 2. Results

### 2.1. Umbilical Cord Blood Characterization Using a Hematology Analyzer

During the selection of UCB units for hematopoietic stem cell transplantation, cell count is one of the main criteria, along with HLA matching between the graft and the patient. The TNC count is determined in freshly collected blood; however, variations in the final product resulting from volume reduction, cryopreservation, and thawing are neither determined nor considered. To characterize the cell populations in UCB units stored at the Public Umbilical Cord Blood in Bogotá, eleven cryopreserved UCB units for transplantation were selected from the inventory. General information about these units is provided in Table 1.

Three stages were selected to characterize variations in the proportion of different cell populations: fresh, after volume reduction (VR), and after thawing. In line with common practice, the proportions of granulocytes, monocytes, and lymphocytes at these three stages were compared using a hematology analyzer in these three moments using a two-way ANOVA analysis (Figure 1). No significant differences were observed in any of these populations across the three stages. According to the data from the hematology analyzer, VR and thawing of the 11 UCB units did not alter the proportions of granulocytes, monocytes, or lymphocytes, which are the primary cell populations in the samples, despite variations in total cell count throughout the process. Because further characterization of cell populations with potential application is required in other types of therapies, these data were validated through flow cytometry.

### 2.2. Umbilical Cord Blood Immunophenotypic Characterization

Flow cytometry is a valuable method for determining the phenotype of hematologic cell populations. However, without proper standardization, results can significantly vary based on the instrument, antibodies, fluorochromes, and data acquisition setup [12]. For this study, the instrument setup, sample preparation, antibody panels, and data analysis followed the EuroFlow consortium’s standardized protocol [13]. The acute leukemia orientation tube (ALOT) was selected as it provides efficient characterization of immature hematopoietic cell populations. Biomarker panels for T cell, B cell, granulocyte, and monocyte lineages were selected to quantify these populations in UCB units, as their isolation could be useful to implement other types of therapies (Appendix A). Immunophenotypes for each population as well as the differentiation state are detailed in the Appendix A. A representative image of the gating strategy for analysis performed following the EuroFlow protocol is shown in Figure 2 for fresh cells, Figure 3 for UCB cells after volume reduction (VR), and Figure 4 for UCB cells after thawing (AT).

Using a two-way ANOVA test, we determined the relative proportion change of granulocytes, monocytes, T cells, B cells, and NK cells characterized using EuroFlow in the 11 UCB units through the processing steps mentioned earlier, as shown in Figure 5.

There were statistically significant differences in the granulocyte and T cell populations when comparing the after-thawing samples with both fresh samples and after VR; the granulocytes population ratio decreased, while the T cells ratio increased after thawing compared with the other two stages (*p* > 0.005). Monocytes, B cells, and NK cells did not exhibit significant variation during the process. Due to the T cells’ proportion in the unit increasing after thawing, a further evaluation of T cell subsets was performed using the total T cells tube. Double negative (CD4^−^ CD8^−^), double positive (CD4^+^ CD8^+^), and CD4^+^ and CD8^+^ T lymphocytes were compared in the three stages of cord blood processing using a two-way ANOVA test (Figure 5). There was only a statistically significant decrease in the CD4^+^ cell population after thawing compared with the after-VR phase. However, the analyzed subpopulations do not seem to be entirely responsible for the total T cell change after thawing.

Some primitive and immature myeloid cell populations were also compared during the three stages to verify their presence and changes in UCB. Myeloid CD34^+^ cells, myelocytes, metamyelocytes, band cells (Figure 6), CD4^−^CD8^−^, and transitional B cells (Figure 6) were compared through two-way ANOVA. Only metamyelocytes and band cells exhibited a statistically significant difference between fresh and after thawing and after VR compared to after thawing. After thawing, there is a critical decrease of this population, although the total percentages are, as expected, considerably lower than the other analyzed populations, as may be observed with the scale in the graph.

### 2.3. Absolute Cell Count per Population Determined Through Flow Cytometry

Because our main goal was to determine whether it would be useful to isolate cell populations from the cryopreserved UCB units with potential use in therapies different from classical stem cell transplantation, we extrapolated the cell count of these populations in the three stages. During processing, the total cell count was obtained using two different methods: a hematology analyzer and flow cytometry using the standardized ISHAGE protocol [14] after volume reduction and after thawing. The ISHAGE protocol also provides the cell density of CD45^+^ and CD34^+^ cells, as well as their viability. However, fresh samples’ cell count is only determined using the hematology analyzer. Therefore, our cell count extrapolation was initially obtained as dual-platform for the three stages and as single-platform for the after-VR and after-thawing stages [15]. For the two-platform comparison, the cell density of white blood cells from the hematology analyzer (cells/µL) was used as the reference to calculate the total cell count of populations according to the proportion obtained per flow cytometry with the EuroFlow method in the principal cell populations, granulocytes, monocytes, T cells, B cells, and NK cells, as shown in Figure 7. Because the total count of granulocytes overcomes the other cell populations, an additional analysis excluding this population was performed. The median of granulocytes is around 1 million cells, with an important decrease after thawing, reaching less than 400,000 cells. Meanwhile, T cells are the second highest population, with approximately 300,000 total cells in the fresh and after-VR stages, exhibiting an apparent increase in total cells due to the proportional increase observed before. Monocytes, B cells, and NK cells, as observed previously, remain stable in proportion, with less than 200,000 total cells at all stages.

As mentioned previously, the single-platform measurement was only obtained for the after-VR and after-thawing stages. In both cases, the total CD45^+^ cell population and the number of cells per population were calculated based on the proportions obtained through EuroFlow. In this case, the total numbers are two orders of magnitude higher than those obtained using the hematology analyzer. The analysis excluding granulocytes was also carried out in this case, as shown in Figure 8.

According to the information obtained from the single-platform evaluation, the UCB units were barely around 7.5 × 10^8^ granulocytes, with a significant decrease after thawing not only for this population but also for the others evaluated. T cells are, as expected, the second highest cell population, with approximately 2.5 × 10^8^ after volume reduction and exhibiting a decrease after thawing. For the remaining cell populations, the behavior is very similar.

The CD34^+^ cell population, which is considered one of the most important for UCB transplantation as it leads to bone marrow repopulation, was also evaluated through a single-platform method only after VR and after thawing using the ISHAGE method. The 11 evaluated units had a mean of 9 × 10^6^ cells with 99% viability after VR, but they exhibited a critical reduction after thawing, reaching a mean of 182,272 total cells and a viability of 57.82%. Regarding the CD34^+^ myeloid cells, the total cell count extrapolated using the two-platform method using Euroflow shows a mean of 4945 total cells in fresh samples (0.27% of the total cell populations evaluated), a slight proportion increase, as expected, after VR, reaching 5795 cells (0.33% of the cell populations), with the trend continuing after thawing (5416 cells, 0.43% of the total cell populations). Appendix A shows the three different conditions evaluated (fresh, after VR, and after thawing) of one representative experiment (Donor N°5 in Table 1) using EuroFlow panels, demonstrating that through this gating strategy, CD34^+^ cells have a positive expression of CD13, CD45 (dim), and HLA-DR and a positive but heterogeneous expression of CD117^het^ [16,17]. These total cell counts are considerably lower compared with the previously mentioned cell populations, which was expected, as the proportion is also quite low in UCB. However, if properly handled and isolated, these could be important cell numbers to use clinically and perform in vitro experimentation, which might help to understand the biology of these populations and their possible uses when isolated or used with other blood cells.

There is an important difference between the cell count extrapolated from a double platform and a single platform. As it has been mentioned earlier, the two-platform system usually exhibits more bias and inaccuracy due to the differences between both instruments, which could lead to an underestimated true cell count.

### 2.4. Viability and Clonogenicity Assays

Assessing the viability of UCB cells is a critical step in predicting the performance of these units, and it is also a standard procedure conducted prior to the distribution of UCB units and HPSC transplantation. To this end, a fragment containing a representative sample of cells from each UCB unit was thawed and analyzed for viability and clonogenicity. Viability was assessed following the ISHAGE protocol [14] and using the clonogenicity efficiency assay (eClone), which is also based on the number of CD34^+^/7AAD^−^ cells measured in the sample and able to induce colony formation [18]. Notably, this clonogenicity efficiency assay, also named eClone, performed on thawed cells has emerged as a more reliable predictor of favorable transplantation outcomes, particularly when its value exceeds 20% [19]. The eClone results from thawed fragments of clinical units listed in our transplantation registry and shown in the present study demonstrated adequate viability, with an average eClone of 31%, ranging from 7.7% to 44.3% (Appendix A); this is only slightly higher than the average of 88 UCB units that have been already distributed for HSPC transplantation from our clinical bank. However, as previously reported, the eClone values, as well as the percentages of viable CD45^+^ and CD34^+^ cells obtained from the unit-attached fragments, do not consistently reflect the overall potency of the entire UCB bag; in general, they show lower numbers [20]. Additionally, these results are highly sensitive to the specific thawing protocol employed, which we have validated previously [18].

Then, we conducted a viability assay using Annexin V and 7-AAD on thawed cells from UCB bags that were not currently listed for transplantation. This approach enabled us to evaluate not only the percentages of necrotic or late apoptotic cells but also those undergoing early apoptosis due to thawing stress. As shown in Figure 9, the viability of four different UCB cell units ranged from 53.7% to 64.8%, while the percentage of cells in early apoptosis ranged from 9.81% to 15.2%. The proportion of cells in late apoptosis or necrosis ranged from 25.4% to 34.42%, as seen in Appendix A. These results confirm the variability of UCB cells in cellular resistance to thawing process stress and the preservation of HSPC potency in these UCB units.

### 2.5. Functional Assays for Cord-Blood-Isolated Cells

Vascular transmigration is a main function of all blood cells. White blood cells migrate when the endothelium is activated, and it expresses molecules, such as VCAM1 or ICAM1, and chemokines, such as CXCL12. This activation could be triggered by the stimulation of vascular endothelial cells performed by granulocytes, monocytes, lymphocytes, or even the stromal cells that surround the vascular tissue. The migration capacity of HPSC cells in mobilized peripheral blood grafts used in transplantation has been shown to be related to clinical outcomes, as bone marrow vascular cells are stimulated after conditioning regimes, participating in HSPC cells’ homing process [21,22]. Therefore, we tested the induced migration ability in thawed mononuclear cells from cord blood donated by different individuals using a flow cytometry gating strategy to classify transmigrated cells with higher FSC-A and SSC-A (P2, monocyte-like cells) from the ones with lower FSC-A and SSC-A (P3, lymphocyte-like cells) and calculating the fold migration induction towards VCAM1 and CXCL12 over spontaneous migration (Figure 10).

We focused on lymphocyte- and monocyte-like cells as these populations exhibited migration induced by VCAM1/CXCL12 stimuli, as shown in Figure 11. Figure 11A shows the FSC-A vs. SSC-A gating strategy used in a representative experiment for each donor. Figure 11B shows a density plot of migrated cells from one donor, where two different cellular populations based on FSC-A and SSC-A parameters were found. Analysis of density plots from different donors showed that the P2 population’s migration is increased in some donors (shown in Figure 11A, below the dotted line) while the P3 population’s migration induction is relatively higher in the plots from donors shown above the dotted line and classified as Profile 1, included in Figure 11C. When analyzing the migratory index for each donor, we could identify at least two profiles: those in which response towards VCAM1 and CXCL12 is mainly driven by lymphocyte-like cells and almost non-monocyte-like cells (Profile 1), and those in which migration is predominantly of the monocyte-like population, with lymphocyte-like cells barely induced by migratory stimuli (Profile 2) (Figure 11D and Figure 11E, respectively). For all donors from Profile 1, the migration index is higher than 15 in the lymphocyte-like cellular population, while in the monocyte-like cellular population, this index is less than 2.5. On the other hand, the cells from Donor Profile 2 had a lymphocyte-like induced migration index of less than 2 and a migratory index for monocyte-like cells reaching 10.

Cell migration capacity is not a standardized test used as quality control in UCB candidates for cellular therapies. We demonstrated here that after the thawing stage, there is significant donor variation in the migration capacity of UCB cell populations. Furthermore, we identified two donor profiles with a differential migratory trend between the evaluated populations, which might be important for different applications in cellular therapy, including hematopoietic transplantation. For the latter, clonogenic assays are being performed to test the functionality of these cells that, in our experiments with clinical units, are correlated with CD34^+^/CD45^+^ cells’ viability [18].

We confirmed that the CD45^+^ cells’ viability after thawing was very similar for all donors (>80%), and the proportion of the P2 and P3 populations before migration was also similar (Appendix A). Notably, all donor cells that migrated in the CXCL12/VCAM1 induction assays were able to form colonies (Figure 12A), confirming the presence of viable and proliferative HSPC. When comparing the number of CFUs between the migrated cells from donors classified as Profile 1 and Profile 2, no differences were found (Figure 12B). Interestingly, when normalizing the CFU count according to the number of events acquired through flow cytometry, there is a significant difference between groups (*p* = 0.03, one-tailed Mann–Whitney test), as shown in Figure 12C. The range of percentages of CFU found in the latter assays are more correlated with the percentages of HSPC cells present in the mononuclear fraction of all sources of CD34^+^, which varies between 0.1% and 4%. These differential migratory phenotypes in specific cellular populations between UCB donors could translate in vivo as a critical cellular function and may benefit different applications in advanced cellular therapies.

## 3. Discussion

The increasing usage of haploidentical transplant worldwide has raised several questions regarding the future of public UCB banks, which have acquired important inventories and expertise related not only to collection, processing, cryopreservation, and thawing of UCB but also graft manipulation and quality assessments [7,8,23]. In our experience, the rate of UCB usage is around 7% of the total inventory, but numerous potential uses for the stored UCB units and their different cell populations have been proposed so far, such as iPSC, platelet concentrates, macrophages’ isolation, dendritic cells’ isolation, CAR-NK, cytotoxic T cell manufacturing, and monocytes, in addition to using the cord and placental tissues as a source of mesenchymal stem cells [7,8,9,23,24,25]. These could also turn into important possibilities for UCB units that are discarded for hematological transplant, as reported in dendritic cell generation from this CD34^+^-cell-enriched source [26]. The Public Umbilical Cord Blood Bank in Bogotá, Colombia has an inventory of 1272 units available for transplant and around 3000 frozen units that cannot be used for hematological transplant due to low total cell dose, low CD34^+^ or CD45^+^ cell viability, or high erythroblast content. Because all of these UCB units were screened according to inclusion and exclusion criteria established for banking, characterized in terms of cell content, HLA typing, microbiological tests, CD34^+^ and CD45^+^ cell number, and viability and serology tested for blood-transmitted infectious diseases, such as human immune-deficiency virus (VIH), hepatitis B virus (HBV), hepatitis C virus (HCV), toxoplasma, cytomegalovirus, and syphilis, both in cord blood and maternal serum, the cells that can be obtained from these units could be eligible for other clinical purposes. To determine which specific cell populations can be found in our stored UCB units and in which proportion, we used the standardized flow cytometry protocol EuroFlow to characterize primitive, granulocytic, monocytic, and lymphocytic cell populations in fresh blood after VR and after thawing. The analyzed populations include those previously mentioned, which have been explored by research groups in different UCB banks and can potentially be isolated after thawing for application in cell therapies. Also, the cell count per population was determined using both a two-platform system with a hematology analyzer—which is the typical instrument to determine if the cell dose is appropriate for transplant—and a single platform with the ISHAGE protocol.

Performing cell analysis of UCB units using flow cytometry allows for higher sensitivity to identify hematopoietic cells, including primitive populations, such as myeloid CD34^+^ cells, mature cells with higher CD45^+^ expression, such as neutrophilic granulocytes, monocytes, and lymphocyte subpopulations, such as T lymphocytes, B lymphocytes, and NK cells [27,28,29]. Additional primitive populations, such as myelocytes, metamyelocytes, and bands, were identified using this methodology. The main cell populations, such as granulocytes, monocytes, and lymphocytes, did not exhibit any variation during the three processing stages evaluated when measured using the hematology analyzer. However, when dissecting these populations using immunophenotypes in the standardized EuroFlow protocol, there were statistically significant differences in granulocyte and T cell populations. In both cases, fresh samples and samples after VR are similar, but the granulocyte count decreases significantly after thawing, as expected, while T cells increase in proportion within the unit. We did not find any T cell subpopulation driving this increase. However, given the immunological properties of UCB T cells previously reported to modulate the graft vs. host and graft vs. leukemia response out of the transplantation context, this could be an important target population for research in our frozen units [9,27]. Another important finding regarding cell count is that the two-platform setting (extrapolation from EuroFlow proportions in cell count obtained from the hematology analyzer) has important differences compared to the one-platform setting. While in the first the mean T cells found in the 11 samples were 3.1 × 10^5^, the latter has a mean of 2.6 × 10^8^ after volume reduction. This raises an important concern regarding the accuracy of cell dose measurements in UCB banks when obtained from hematology analyzers. If the cell count from the one-platform setting is correct, there is an important number of T cells that could be isolated and employed for cell therapy after volume reduction and even after thawing.

Primitive myeloid cells, such as myeloid CD34^+^ cells, myelocytes, metamyelocytes, and band cells, and immature populations, such as CD4^−^CD8^−^ and transitional B cells, were found in small proportions in all units. However, the only significant variation in the three processing stages regarded metamyelocytes and band cells, with an important decrease after thawing. However, if these cells can be isolated after thawing with acceptable viability, it could be possible to perform cell differentiation in vitro, which can contribute to regenerative medicine.

Regarding the cell count, as previously mentioned, there is a huge difference between the two employed settings. Previous publications have attributed more accuracy to the single platform, as it is based on a single instrument and type of measurement to determine the total cell count in a sample [15]. Furthermore, both flow cytometry methods performed (ISHAGE and EuroFlow) are standardized for clinical purposes, which suggests a more reliable result [12,29,30,31,32,33]. Therefore, according to the obtained data, a mean of 5 × 10^6^ monocytes, 1.8 × 10^7^ T cells, 4.7 × 10^6^ B cells, and 3.7 × 10^6^ NK cells could be isolated and used for cell therapy after thawing from the UCB units evaluated.

Recently, UCB T cells were preselected using the CD62L marker for a machine-based manufacturing process for CAR-T production that could help widen access to CAR-T cells [29]; CD62 is an adhesion marker expressed in endothelial cells, which can also direct the migratory phenotype of T cells. Georgiadis et al. demonstrated the feasibility of producing and cryopreserving enough universal CAR19-T cells, which could offer an alternative to autologous advanced cell therapies [34].

As previously mentioned, the migratory behavior of CD34^+^/CD45^+^ cells from a mobilized peripheral blood graft has been shown to correlate with lower neutrophil engraftment times [17]. Voermans and collaborators tested the migration of isolated CD34^+^/CD45^+^ cells through fibronectin or the CXCL12 chemotactic gradient and showed an inverse correlation between migration and neutrophil engraftment time in hematopoietic stem cell transplantation. Zhao and collaborators studied the single-cell transcriptome of UCB mononuclear cells and identified two subpopulations related to progenitor cells in UCB, which is congruent with the two profiles identified in the migration assays mentioned before [35]. We could speculate that the migratory behavior towards VCAM1 and CXCL12 observed in our study could lead to faster engraftment of hematopoietic cells in UCB donors classified as Profile 1. Our experiments demonstrate that these migratory cells generate a relatively higher number of hematopoietic colonies, reinforcing their potential advantage. However, it is important to remember that HPSC transplantation is performed with biological sources that contain mostly differentiated cells (in contrast to the smaller percentage of HSPC, between 0.3% and 4%), which would have a potential role in the physiology of the transplantation, as demonstrated by different authors [36,37]. While we acknowledge that this represents a reductionist view of a highly complex process, it aligns with well-established findings; blocking the CXCL12/CXCR4 and VCAM1/VLA4 pathways in various models, including humans, consistently mobilizes diverse hematopoietic cell types—most notably HSPCs—from the bone marrow into peripheral blood [38]. All of these results suggest the need to develop efficient tests that might help to define the potential of the UCB that can be employed in cell therapy. The intrinsic variability of the cellular populations or the functional assays can be exploited in cell banks for allogeneic donors. It would also be interesting to assess how these profiles may impact clinical outcomes of hematopoietic progenitor and stem cell transplantation or the development of specific advanced therapies.

## 4. Materials and Methods

### 4.1. UCB Collection, Processing, Cryopreservation, and Thawing Process

UCB units of eleven donors were selected through standard collection processes at the Public Umbilical Cord Blood Bank in Bogotá, Colombia (Instituto Distrital de Ciencia, Tecnología e Innovación IDCBIS) in public hospitals in Bogotá, as previously described [11]. Maternal and partner clinical histories were obtained, and all donors signed an informed consent form reviewed and approved by the Ethical Committee of the District Secretary of Health. Collection was performed using the in utero/ex utero method by professional nurses from the UCB bank, previously described by our group [11].

Cord blood was collected in bags containing citrate–phosphate–dextrose solution and transported to the laboratory. The total nucleated cell (TNC) count was determined using a hematology analyzer (Sysmex XN-1000™). Flow cytometry characterization was performed according to the standardized ISHAGE [14] and EuroFlow protocols, which are further detailed in subsequent sections. Volume reduction was then performed using the closed system Sepax (Biosafe S.A., Kuopio, Finland) with hydroxyethyl starch (HES) 6%. After processing, the TNC count was re-evaluated using the hematology analyzer, and flow cytometry was repeated, as previously mentioned.

All UCB samples were cryopreserved using DMSO CryoSure-DEX4 using a controlled freezing curve. The samples were stored in vapor-phase nitrogen below −180 °C. Each unit included the standard three segments for quality assessments and an additional unattached 5 cm tubing segment reserved for post-thawing flow cytometry characterization.

Thawing was performed as previously described by our group [18]. Briefly, the sample was removed from the liquid nitrogen container and thawed at room temperature, followed by a final dilution of 1:8 in thawing media (0.6% HES, 4.2% bovine serum albumin (BSA) phosphate-buffered saline (PBS 1X)).

### 4.2. Cell Characterization Using the EuroFlow Protocol

Instrument setup was performed based on previous reports for the EuroFlow protocol [30]. A FACSCanto II cytometer was used to capture the events for all experiments while using BD FACSDiva^TM^ software version 6.1.3 to monitor the device’s performance and making adjustments for 8 fluorochromes (V450, V500c, FITC, PE, PERCPCY5.5, PECY7, APC, and APCH7) using OneComp eBeads^TM^ compensation beads. For staining, 400 µL of fresh blood samples, after volume reduction, or after thawing was used. The sample was centrifuged and resuspended in the same volume of PBS 1X + fetal bovine Serum (FBS) 0.5% + Sodium Azide 0.09%, and 50 µL was dispensed in each of the 7 tubes, 2 controls (cell membrane and cytoplasm staining) and 5 tubes with antibodies for identification and quantification of different cell populations, distributed as described in Appendix A into five main groups: immature hematopoietic cells (Appendix A), T lymphocytes (Appendix A), B lymphocytes (Appendix A), granulocytic lineage (Appendix A), and monocytic lineage (Appendix A). Analysis was performed with Infinicyt™ version 2.0. (Cytognos SL, Santa Marta de Tormes, Spain).

### 4.3. Migration Assays

For the migration assays, mononuclear cells from nine different donors from non-clinical UCB units were used. Migration assays were performed using a 6.5 mm diameter 5 µm pore Transwell (Costar, Kennebunk, ME, USA). Transwell filters were coated with 10 µg/mL of VCAM1 (Sigma-Aldrich, Life Science, Saint Louis, MO, USA) in PBS 1X at 37 °C overnight. After two washes with PBS 1X, the Transwell membranes were activated with a migration buffer (RPMI 1640 medium supplemented with 2% BSA) for 30 min at 37 °C.

UCB mononuclear cells from different donors were thawed using a slight modification of our validated protocol [18], and flow cytometry characterization was performed according to the standardized ISHAGE [14] (Appendix A). Then, 3 × 10^5^ cells were added to the upper compartment in 100 µL of migration buffer. Spontaneous migration was assessed by filling the lower compartment of the Transwell with 600 µL of migration buffer, while induced migration was assessed after the addition of 50 ng/mL of CXCL-12 (R&D Systems^Ⓡ^, Minneapolis, MN, USA) diluted in the migration buffer. The Transwell plates were incubated for 3 h at 37 °C with 5% CO_2_ [39].

Cells that migrated to the lower compartment were collected, counted, and analyzed using flow cytometry to calculate the induced migration normalized with the spontaneous migration. The flow cytometry conditions were as follows: threshold: 200; flow rate: low; stopping time: 60 s; events recorded: 1,000,000; voltages set to FSC:260/SCC:375. Data acquisition and analysis were performed using FACSDiva software version 6.1.3 (BD Biosciences, Franklin Lakes, NJ, USA).

The migration index was calculated as the ratio of the number of induced migration (VCAM1/CXCL12) events and the number of spontaneous migration events.

To test the clonogenic potency of migrated cells, 300 µL of the content recovered in the lower chamber of the Transwell plate was seeded in Methocult (Stem Cell Technologies, Vancouver, BC, Canada), as described below.

### 4.4. Clonogenic Assays

To evaluate functional viability of progenitor cells, we determined the proliferative potential of these cells through clonogenic efficiency (eClone; percentage of effective colonies from a known number of cultured CD34^+^ cells) [18] by counting the overall number of colonies forming units (CFU). Flow cytometry was performed as described previously, to count the number of CD34^+^ viable cells, as well as Total Nucleated Cell (TNC) count. For clonogenic culture assays, 150 CD34^+^ cells/mL were seeded in media (Methocult, Stem Cell Technologies, Vancouver, BC, Canada) per well, with two wells per UCB unit and cultured during 14 days at 37 °C and 5% of CO_2_. eClone was calculated as the proportion of colonies formed compared to the initial 150 CD34^+^ cells. Unit colony formation assays were performed after CXCL12/VCAM-1 induced migration assays were performed, using 300 µL of the volume contained in the lower chamber of the Transwell plates in 1.1 mL of Methocult (Stem Cell Technologies, Vancouver, BC, Canada). Two wells per donor were cultured during 14 days at 37 °C and 5% of CO_2_. Normalization of the percentage of CFU was performed using the total events seeded (counted by flow cytometry) taken as the 100% of cells in the migration assays.

### 4.5. Apoptosis/Necrosis Detection Assay

The GFP-CERTIFIED^®^ Apoptosis/Necrosis Detection Kit (ENZ-51002, ENZO Life Sciences, Farmingdale, NY, USA) was used to assess viable, early apoptotic, and late apoptotic cells in thawed UCB samples. Cells from whole cryopreserved CBU bags were thawed in a solution of 0.6% HES + 4.2% HSA at a concentration of 1 × 10^6^ cells/mL. For apoptosis control, 2 μM of staurosporine was added, and the cells were incubated for 8 h at 37 °C. For necrosis control, the cells were incubated at 56 °C for 30 min in 1 mL of 1X PBS. Viable cells were maintained in 1X PBS without any additional treatment (controls are shown in Appendix A). Four UCB units from different donors were thawed. The samples were centrifuged at 400× *g* for 5 min, the supernatant was removed, and the cells were washed with 1X PBS. A staining solution containing 5 μL of Annexin V EnzoGold reagent and 5 μL of 7-AAD reagent was added to each sample. The samples were incubated at room temperature (RT) for 15 min while protected from light. Finally, the cells were acquired using a FACSCanto II cytometer (BD) and analyzed with FlowJo software version 10.6.2.

### 4.6. Statistical Methods

All statistical analyses were performed using GraphPad Prism software version 9.0.1. Comparisons of either relative or absolute values were performed, including the 11 units analyzed in the three moments of processing: fresh blood, after volume reduction (VR), and after thawing. Two-way ANOVA analysis was performed in all analyses. Mann–Whitney tests were performed for CFU and normalized CFU assays by comparing cellular migratory Profile 1 and Profile 2.

## 5. Conclusions

In conclusion, UCB units intended for therapeutic purposes should meet specific quality control criteria. This research proposed analyzing the proportions of cell populations in UCB units, as this could provide a potential advantage in predicting graft-versus-host disease prognosis and engraftment outcomes due to the modulation of each cell population within the altered medullary niche of the host [9,23,34]. Furthermore, this study highlighted the importance of flow cytometry analysis compared to hematology counts, as it revealed the underestimation of cellular loss post-thawing, particularly in granulocytes, and the overestimation of viability for some populations. Specifically, in the context of UCB transplantation, our results do not support the exclusion of any specific donor profile. While we observed that donors classified as Profile 1 exhibit a relatively higher proportion of HSPCs migrating toward CXCL12 and VCAM1, donor selection should also consider other critical factors, such as the viable TNC/kg dose, the CD34^+^/kg count, and patient compatibility. However, these distinct migration patterns could serve as useful criteria for advanced therapies involving monocytes or lymphocytes derived from UCB cells. Selecting specific donor profiles for these therapies may enhance outcomes. We conclude that the interindividual phenotypic and function variability of banked UCB cells is extensive and highly dependent on the methodologies used for assessment. Thus, this variability should be explored and leveraged rationally based on the intended therapeutic application, including HSPC transplantation or advanced therapies derived from UCB cells.

## Figures and Tables

**Figure 1 ijms-26-01208-f001:**
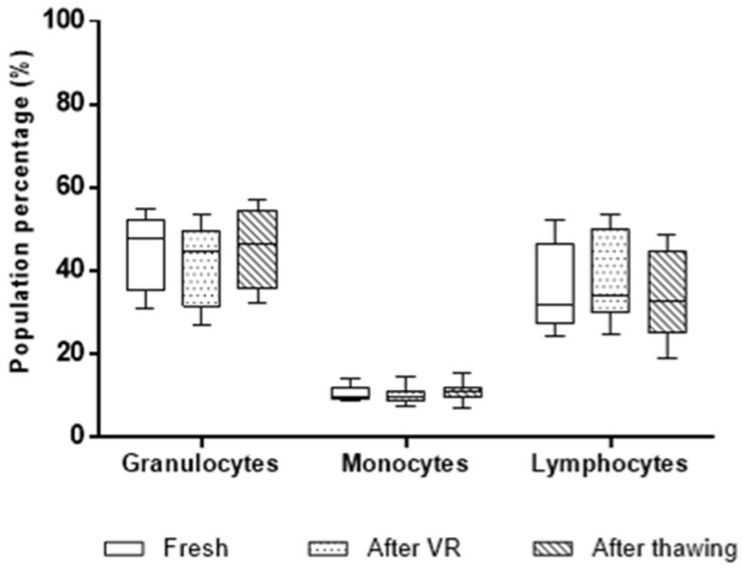
Box-and-whisker plots with the proportions of cell populations (granulocytes, monocytes, and lymphocytes) in the 11 analyzed UCB units at the three processing stages: fresh, after volume reduction (VR), and after thawing. A two-way ANOVA analysis revealed no significant differences in cell population proportions between the stages.

**Figure 2 ijms-26-01208-f002:**
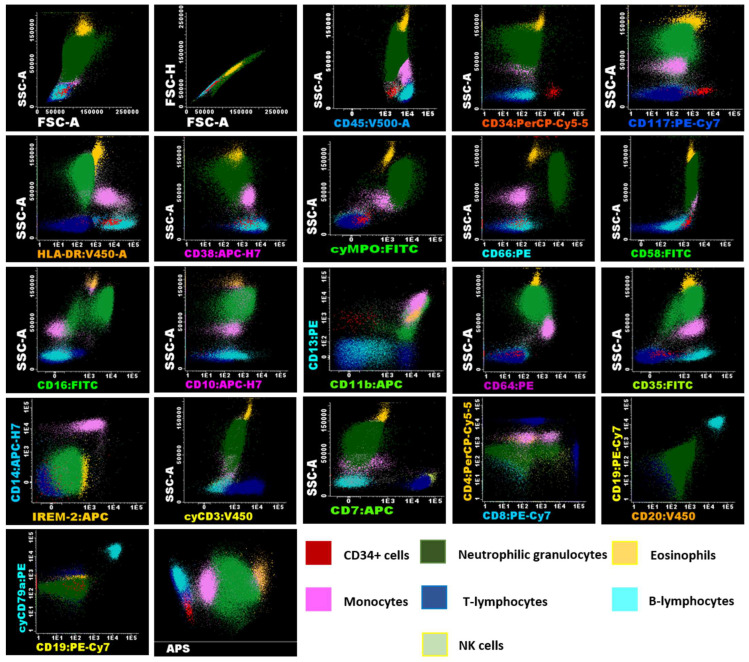
Flow cytometry analysis of umbilical cord blood samples using EuroFlow panels of fresh UCB cells. Immunophenotypic analysis was conducted using Infinicyt™ version 2.0. (Cytognos SL) software using fresh cells of Donor N° 5 (see Table 1), displaying the expression of various markers analyzed in the antibody panels, which facilitates the identification of myeloid and lymphoid lineages in the samples. Identification of these populations was based on their characteristics, including SSC, FSC, the differential expression of CD45, and the presence or absence of other panel markers. The analysis revealed that CD34^+^ cells (shown in red), characterized by low SSC and FSC, expressed CD117^+het^/HLA-DR^+^/CD45^+^ dim/CD38^+^/CD58^+^/CD13^+^. Neutrophil granulocytes (shown in green), exhibiting high SSC and FSC, were CD45^+^/cyMPO^+^/CD66c^+^/CD58^+^/CD16^+^/CD10^+^/CD13^+^/CD11b^+^/CD64^+^dim/CD35^+^. Eosinophils (shown in orange), also with high SSC and FSC, were characteristically CD45^+^/CD58^+^/CD13^+^/CD11b^+^. The immunophenotype of monocytes (shown in pink), characterized by intermediate SSC and FSC, was CD45^+^/HLA-DR^+^/CD38^+^/cyMPO^+dim^/CD58^+^/CD13^+^/CD11b^+^/CD64^++^/CD35^+^/CD14^+^/IREM-2^+^. In the panels used, lymphocyte subpopulations (low SSC and FSC/CD45^+^) were characterized separately, including CD3^+^/cyCD3^+^/CD2^+^/CD7^+^/CD38^+^/CD58^+^ T cells (shown in blue), with CD4^+^ and CD8^+^ subgroups identified. B lymphocytes (shown in cyan) were CD19^+^/CD20^+^/cyCD79a^+^/HLA-DR^+^/CD38^+^/CD35^+^, and NK cells were CD2^+^/CD7^+^/CD38^+^. The analysis also included an APS (Automatic Population Separator) diagram based on principal component analysis (PCA), which clearly demonstrates the automatic n-dimensional separation of the major cell clusters within the sample.

**Figure 3 ijms-26-01208-f003:**
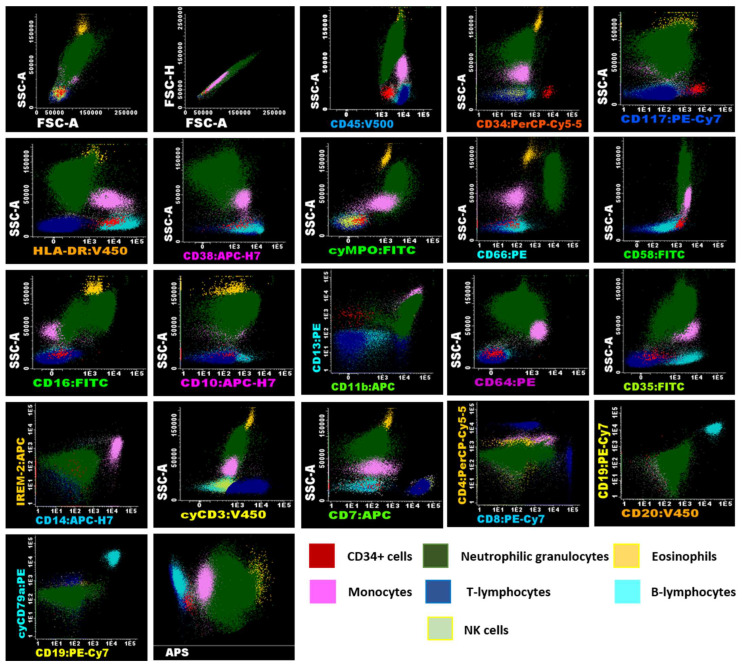
Flow cytometry analysis of umbilical cord blood samples using EuroFlow panels of UCB cells after volume reduction (VR). Immunophenotypic analysis was conducted using Infinicyt™ version 2.0. (Cytognos SL) software using cells from Donor N° 5 (Table 1) after volume reduction (VR), displaying the expression of various markers analyzed in the antibody panels, which facilitates the identification of myeloid and lymphoid lineages in the samples. Identification of these populations was based on their characteristics, including SSC, FSC, the differential expression of CD45, and the presence or absence of other panel markers. The analysis revealed that CD34^+^ cells (shown in red), characterized by low SSC and FSC, expressed CD117^+het^/HLA-DR^+^/CD45^+^ dim/CD38^+^/CD58^+^/CD13^+^. Neutrophil granulocytes (shown in green), exhibiting high SSC and FSC, were CD45^+^/cyMPO^+^/CD66c^+^/CD58^+^/CD16^+^/CD10^+^/CD13^+^/CD11b^+^/CD64^+^dim/CD35^+^. Eosinophils (shown in orange), also with high SSC and FSC, were characteristically CD45^+^/CD58^+^/CD13^+^/CD11b^+^. The immunophenotype of monocytes (shown in pink), characterized by intermediate SSC and FSC, was CD45^+^/HLA-DR^+^/CD38^+^/cyMPO^+dim^/CD58^+^/CD13^+^/CD11b^+^/CD64^++^/CD35^+^/CD14^+^/IREM-2^+^. In the panels used, lymphocyte subpopulations (low SSC and FSC/CD45^+^) were characterized separately, including CD3^+^/cyCD3^+^/CD2^+^/CD7^+^/CD38^+^/CD58^+^ T cells (shown in blue), with CD4^+^ and CD8^+^ subgroups identified. B lymphocytes (shown in cyan) were CD19^+^/CD20^+^/cyCD79a^+^/HLA-DR^+^/CD38^+^/CD35^+^, and NK cells were CD2^+^/CD7^+^/CD38^+^. The analysis also included an APS (Automatic Population Separator) diagram based on principal component analysis (PCA), which clearly demonstrates the automatic n-dimensional separation of the major cell clusters within the sample.

**Figure 4 ijms-26-01208-f004:**
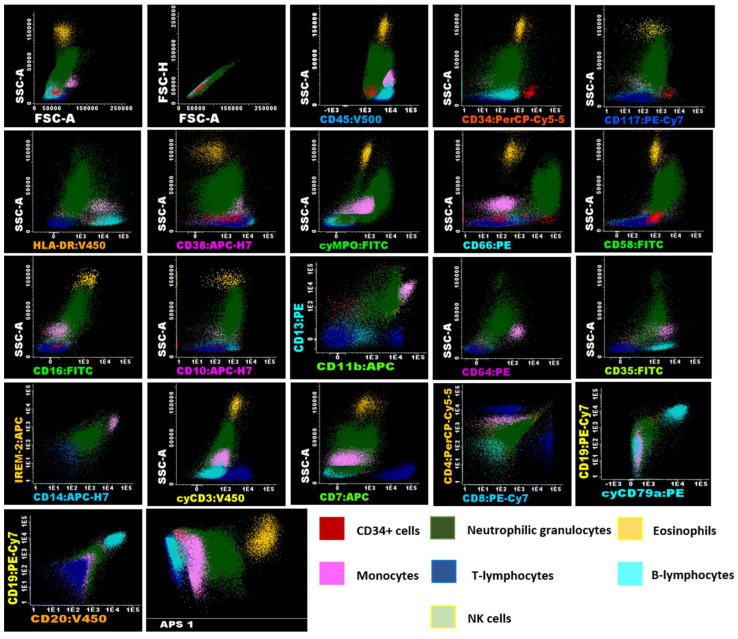
Flow cytometry analysis of umbilical cord blood samples using EuroFlow panels of UCB cells after thawing. Immunophenotypic analysis was conducted using Infinicyt™ version 2.0. (Cytognos SL) software using cells from Donor N°5 (see Table 1) after thawing, displaying the expression of various markers analyzed in the antibody panels, which facilitates the identification of myeloid and lymphoid lineages in the samples. Identification of these populations was based on their characteristics, including SSC, FSC, the differential expression of CD45, and the presence or absence of other panel markers. The analysis revealed that CD34^+^ cells (shown in red), characterized by low SSC and FSC, expressed CD117^+het^/HLA-DR^+^/CD45^+^ dim/CD38^+^/CD58^+^/CD13^+^. Neutrophil granulocytes (shown in green), exhibiting high SSC and FSC, were CD45^+^/cyMPO^+^/CD66c^+^/CD58^+^/CD16^+^/CD10^+^/CD13^+^/CD11b^+^/CD64^+^dim/CD35^+^. Eosinophils (shown in orange), also with high SSC and FSC, were characteristically CD45^+^/CD58^+^/CD13^+^/CD11b^+^. The immunophenotype of monocytes (shown in pink), characterized by intermediate SSC and FSC, was CD45^+^/HLA-DR^+^/CD38^+^/cyMPO^+dim^/CD58^+^/CD13^+^/CD11b^+^/CD64^++^/CD35^+^/CD14^+^/IREM-2^+^. In the panels used, lymphocyte subpopulations (low SSC and FSC/CD45^+^) were characterized separately, including CD3^+^/cyCD3^+^/CD2^+^/CD7^+^/CD38^+^/CD58^+^ T cells (shown in blue), with CD4^+^ and CD8^+^ subgroups identified. B lymphocytes (shown in cyan) were CD19^+^/CD20^+^/cyCD79a^+^/HLA-DR^+^/CD38^+^/CD35^+^, and NK cells were CD2^+^/CD7^+^/CD38^+^. The analysis also included an APS (Automatic Population Separator) diagram based on principal component analysis (PCA), which clearly demonstrates the automatic n-dimensional separation of the major cell clusters within the sample.

**Figure 5 ijms-26-01208-f005:**
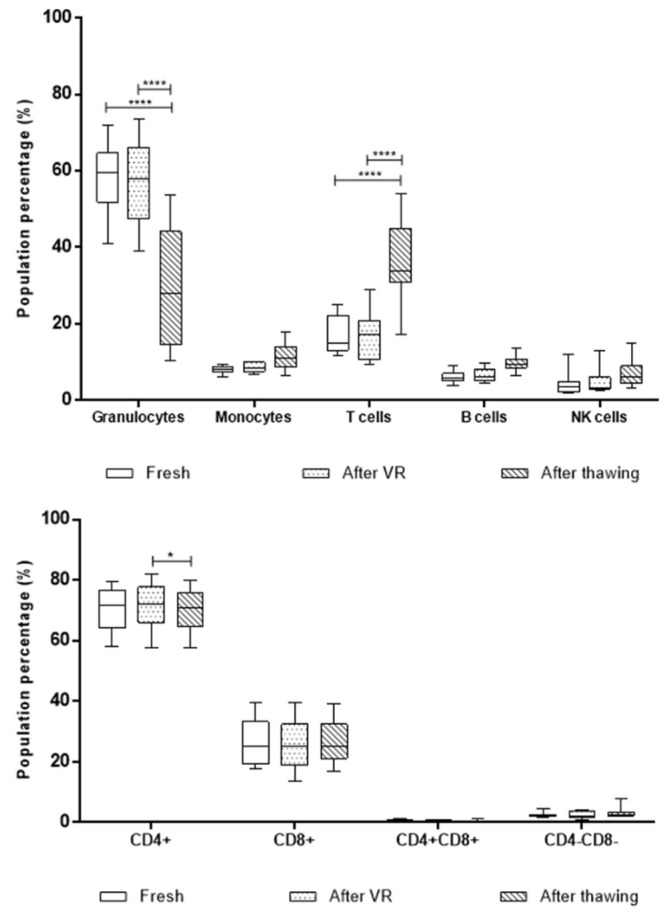
Box-and-whiskers plot of proportion in percentage of different populations in the 11 UCB units analyzed per flow cytometry. Analysis was performed with a two-way ANOVA test. The granulocyte population has significant differences in the three compared stages, fresh, after volume reduction, and after thawing, exhibiting a critical decrease after thawing. T cells’ proportion in the unit shows a significant increase after thawing compared with both fresh and after-VR stages. However, when sectioning the T cell population, the CD4^+^ population exhibits a slight decrease after thawing. (****) corresponds to *p* < 0.001, and (*) *p* ≤ 0.05.

**Figure 6 ijms-26-01208-f006:**
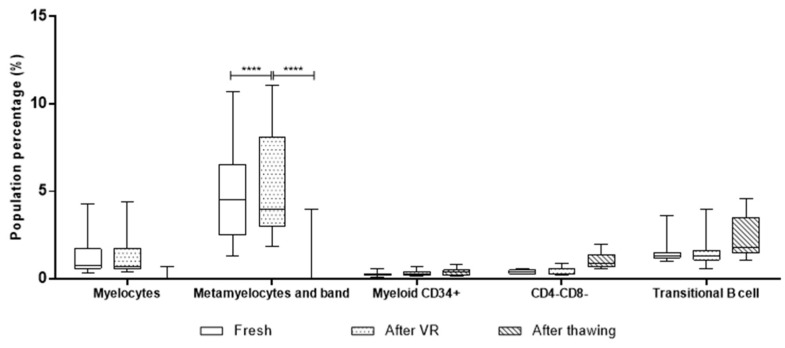
Box-and-whiskers plot with proportion of primitive cell populations compared in the three stages. There were only significant differences for the metamyelocyte and band cell population, which exhibits an important decrease after thawing compared with the fresh and after-VR stage. The remaining population’s proportion remained stable during UCB processing. Box-and-whiskers plot with proportion of immature myeloid cells compared in the three stages. There were no statistically significant differences for these populations during the process. (****) corresponds to *p* < 0.001.

**Figure 7 ijms-26-01208-f007:**
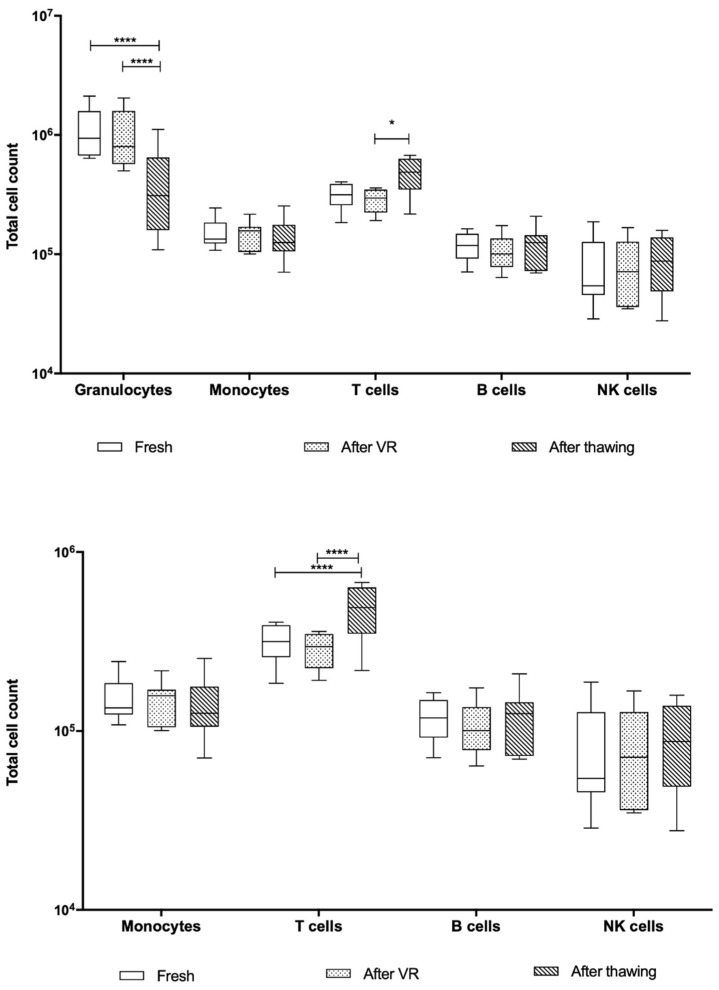
Cell counts per population extrapolated from hematology analyzer data. White blood cell density was used to calculate the total cell number in the UCB unit and, afterwards, to calculate the total cell population based on the proportions determined through flow cytometry. The upper figure includes granulocytes, while the lower figure excludes them to verify the significance of the remaining populations with less than 1 million cells. (****) corresponds to *p* < 0.001, and (*) *p* ≤ 0.05.

**Figure 8 ijms-26-01208-f008:**
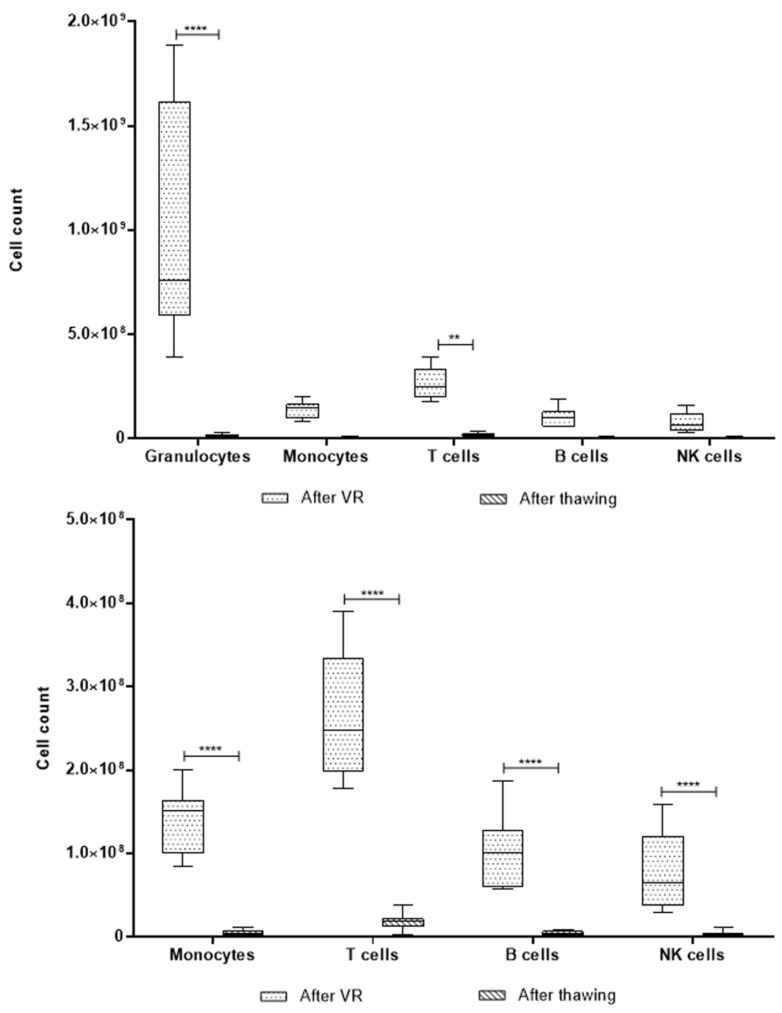
Box-and-whiskers plot of cell count using a single-platform model. All UCB units were analyzed per the ISHAGE protocol to determine the CD45^+^ cell count. The total cell count per population was extrapolated using the total CD45^+^ count and the proportions determined through EuroFlow. (****) corresponds to *p* < 0.001, and (**) *p* < 0.01.

**Figure 9 ijms-26-01208-f009:**
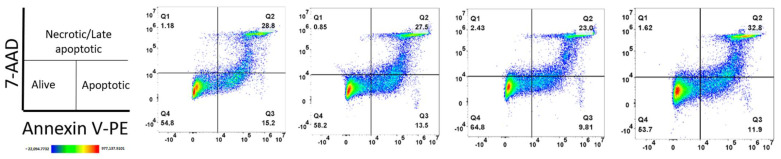
Annexin V/7—AAD viability assay. Thawed cells from four different UCB donors were analyzed. These cells were thawed from 35 mL cryopreserved bags that were deemed unsuitable for clinical transplantation. Quadrant gating for the analysis was established using a staurosporine-treated UCB cell control. Each density plot represents data from a different UCB donor.

**Figure 10 ijms-26-01208-f010:**
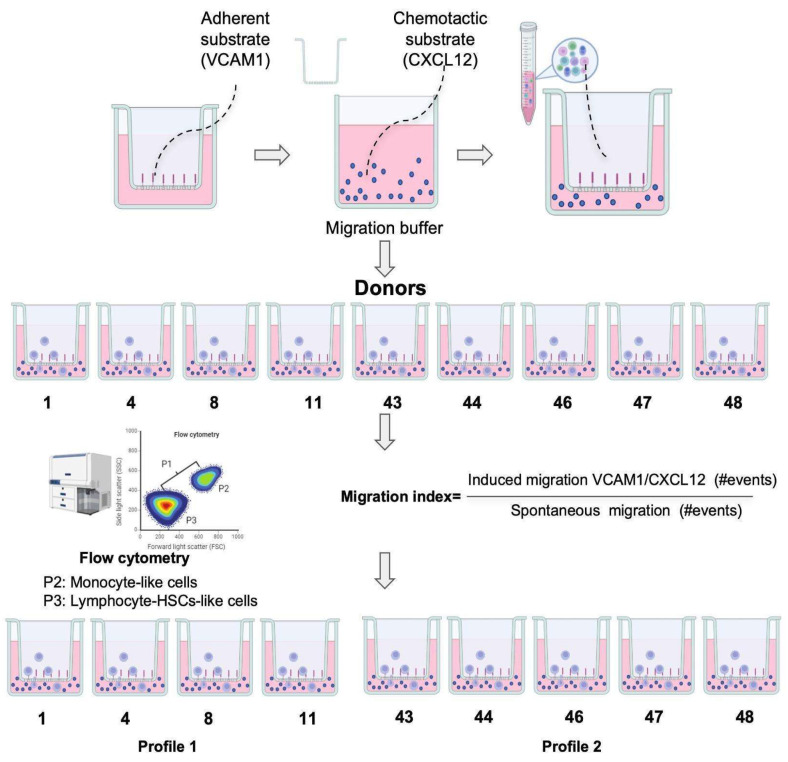
Illustration of the Transwell Assay Method. Each well consists of two chambers. The lower chamber contains a suspension of migration buffer (RPMI 1640 medium supplemented with 2% BSA, shown as pink medium). The upper transwell insert chamber features a polycarbonate membrane coated with 10 µg/mL of the adherent substrate VCAM-1 (represented by red rectangles) and filled with the test chemotactic substrate CXCL12 (depicted as small blue circles). The lower chamber is in contact with a suspension of mononuclear cells (represented by large purple circles), facilitating the migration assay. For the migratory assay using thawed mononuclear cells from different UCB donors, as described in the Materials and Methods section. Isolated UCB mononuclear cells were tested for spontaneous and VCAM1/CXCL12-induced migration in duplicate. The migration index was calculated as the number of events obtained through flow cytometry driven by induced migration with chemotactic stimuli over the number of events obtained with spontaneous migration (without chemotactic stimuli).

**Figure 11 ijms-26-01208-f011:**
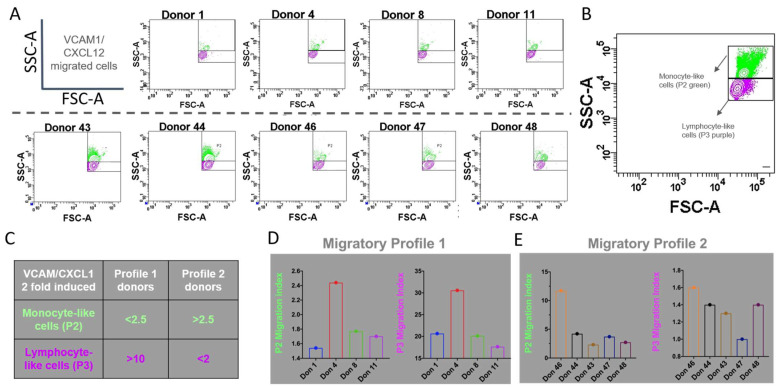
VCAM1/CXCL12 migrated cells from different UCB donors were counted through flow cytometry, as explained in the Methods section. (**A**) Forward Scatter vs. Side Scatter parameters of migrated cells were analyzed for nine donors in duplicate. (**B**) Two populations were identified by volume and complexity: P2 (monocyte-like cells, green) and P3 (lymphocyte-like cells, purple). (**C**) Donor migratory profile classification based on the migration indexes for monocyte-like cells (P2) and lymphocyte-like (P3) populations. (**D**,**E**) Migration indexes for monocyte-like (P2) and lymphocyte-like (P3) cellular populations for donors with Profile 1 (**D**) and donors with Profile 2 (**E**).

**Figure 12 ijms-26-01208-f012:**
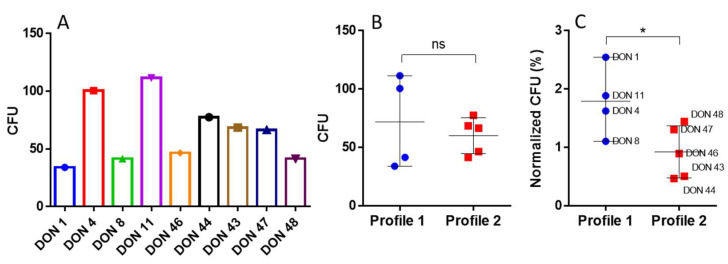
Colony-forming unit numbers of VCAM1/CXCL12 migrated cells from different UCB donors. (**A**). CFUs were analyzed for nine UCB donors. (**B**). Comparison between Profile 1 and Profile 2 donor cells’ ability to form colonies. (**C**). Normalized comparison of the percentage of CFU based on the total events counted through flow cytometry, as explained in the Methods section. (*) indicates *p* < 0.05, as evaluated by Mann Whitney test, ns, not significant.

**Table 1 ijms-26-01208-t001:** General information about the 11 umbilical cord blood units selected for cell population characterization. The data included are the initial blood volume (in mL), the initial total nucleated cell (TNC) count in fresh samples and after volume reduction (VR), the total CD45^+^ and CD34^+^ cell counts after volume reduction, and the viability percentage of CD45^+^ and CD34^+^ cells after volume reduction.

Donor	Initial Volume (mL)	Initial TNCs (×10^8^)	TNCs (×10^8^) After VR	Total CD45^+^ Cells (×10^8^) After VR	CD45^+^ Cells’ Viability (%) After VR	Total CD34^+^ Cells (×10^6^) After VR	CD34^+^ Cells’ Viability (%) After VR
1	146.8	25.9	15.11	20	98.5	9	98.9
2	123.1	16.7	11.3	13.5	98.1	4.6	99.6
3	133.1	15.6	9.86	10	97.1	3.8	98.9
4	86.2	13	11.66	9.5	99.3	1.5	99.3
5	82.2	15.1	14.14	10	NA	1.7	98.4
6	85.7	13.6	12.34	10.8	99.1	5.6	99.5
7	130.9	24.8	15.86	18.4	98.6	3.6	96.9
8	119.7	14.3	9.87	16.3	99.1	11.7	99.1
9	136	23.7	14.72	26.3	97.9	10.1	99
10	136.8	21.3	13.17	25.9	98.2	11.1	98.8
11	165.3	112.6	21.63	21.9	98.6	8.8	98.7

## Data Availability

To request data from this study please contact corresponding author.

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
