# Peer review of "Immunophenotypic and Functional Interindividual Variability in Banked Cord Blood Cells: Insights for Advanced Therapies"

_ijms, 2025, doi:10.3390/ijms26031208_

Round 1
Reviewer 1 Report
Comments and Suggestions for Authors
In this paper, the author organized data from a cord-blood-bank to show the uncontrollable variability of cord blood cells for individual use. To maximize the potential of banked UCB cells for clinical therapies, they focused on UCB cellular populations,viability, and functionality at three processing stages by the means of flow cytometry methods. Significant changes observed in granulocytes and T lymphocytes. Further, by using an in vitro migration assay, the author revealed that thawed UCB cells displayed two distinct migration profiles, lymphocyte-like and monocyte-like cells. Thus, this study concluded to provide a quality control evaluation for cellular property under the procedure of UCB preservation. While the cases they collected for analysis (11) is somewhat in short, data showed some relationship between the property of cell population and function. Though it is short in scientific innovation, overall, this study provided a better understanding for the preservation of UCB cell for further clinical use. A clear shortcoming in this study is there lacks a standardized criteria to select suitable candidates of UCB samples to direct in clinical use. In addition, the analysis methods in this study should also modified.
[Major questions]
1, I do not know how they assay the viability of thawed CD 34+ cells (Table 1)? Since the transplantable HSPCs are enriched in CD34+ population, the author should give more convincible evidence to show the functional viability of these HSPCs, such as by colony assay or suspension culture in vitro, and transplantation ability in vivo.
2, In Figure2, the author showed that CD34+ population were c-kit+ (show in red). It is not the truth. c-kit is not a definitive marker for human HSPC, while it is more significantly expressed on mast cells and their progenitors. Besides, in fresh CB, only about half of CD34+ cells co-express c-kit. The author should correct this improper expression. On the other hand, the author should clearly label the sample No. and stages of three cases in the FACS panel In Figure 2.
3. In Figure 7 and Figure 8, the author showed migration ability of thawed mononuclear cells from different donors. However, the author only used a rough FACS analysis pattern by FSC vs SSC to distinguish lympho/HSC- and mono-like cells. This data is not meaningful, simply because migration under VCAM1 /CXCL12 stimulation could not reflex the true effect of UBC cells in a living body. On the other hand, the proportion of granulocytes in samples, particularly neutrophils and eosinophils, can greatly affect the results. To show the ability of UCB HSPC cell migration, the author should provide more defined and detailed experiment results, such as co-culture with HUVECs or MSCs by trans-well method, and others.
4, In conclusion part, the author should discuss more about establishing a criteria to distinguish the usable UCB samples from those un-usable ones.
Author Response
Authors reply to reviewers:
Reviewer 1:
Comment 1: I do not know how they assay the viability of thawed CD34+ cells (Table 1)? Since the transplantable HSPCs are enriched in CD34+ population, the author should give more convincible evidence to show the functional viability of these HSPCs, such as by colony assay or suspension culture in vitro, and transplantation ability in vivo.
Answer 1:
Thank you for your suggestion regarding the viability of thawed cells, the addition of this data significantly improved our manuscript. We have added a new supplementary table (Table S6) explaining how the ISHAGE protocol was used to assess viability. Furthermore, based on your recommendation, we conducted colony assays to evaluate the functional viability of the cells used in this study. The methods description was included in the new draft version on pages 4 and 5 (lines from 181 to 199) while the results were detailed in a new section, “3.3 Viability and Clonogenicity Assays,” (pages 15 and 16, lines from 474 to 512) which also discusses the stress induced by the thawing process. We apologize for the errors in Table 1. During formatting, the results in the last columns were inadvertently altered. We have corrected the table to present accurate data on CD34+ cell viability and added a red-marked column to highlight the changes.
Comment 2: In Figure2, the author showed that CD34+ population were c-kit+ (show in red). It is not the truth. c-kit is not a definitive marker for human HSPC, while it is more significantly expressed on mast cells and their progenitors. Besides, in fresh CB, only about half of CD34+ cells co-express c-kit. The author should correct this improper expression. On the other hand, the author should clearly label the sample No. and stages of three cases in the FACS panel In Figure 2.
Answer 2:
We appreciate the reviewer’s insightful comments regarding the expression of CD117. Receptor CD117 (c-kit) is indeed expressed in a subpopulation of hematopoietic stem and progenitor cells (HPCs), including CD34+ cells, both in bone marrow and umbilical cord blood. However, its expression is not universal across all CD34+ cell populations and varies depending on tissue type and differentiation stage.
In normal human bone marrow, CD34+ HPSCs and precursor cells express CD117, with variation linked to lineage commitment. For instance, CD34+ precursors committed to the mast cell lineage exhibit a CD34+/CD117hi phenotype, whereas precursors of the granulocytic lineage and various dendritic cell compartments are CD34+/CD117+ (Orfao et al., 2019; Matarraz et al., 2008).
To clarify this statement, we have updated the legend of Figure 2 to accurately describe the analysis of one representative umbilical cord blood sample under fresh conditions, showing the expression of markers analyzed using antibody panels. The analyzed myeloid CD34+ cells display heterogeneous expression of the CD117 marker.
Following your recommendation, we included two new figures (Figures 3 and 4), which present the same data as Figure 2 but under the "after volume reduction" and "after thawing" conditions. Additionally, we added a supplementary figure showing CD117 expression in CD34+ cells across all three experimental conditions (fresh, after VR, and after thawing).
Orfao A, Matarraz S, Pérez-Andrés M, Almeida J, Teodosio C, Berkowska MA, et al. Immunophenotypic dissection of normal hematopoiesis. J Immunol Methods [Internet]. 2019;475(112684):112684. Available from: http://dx.doi.org/10.1016/j.jim.2019.112684
Matarraz S, López A, Barrena S, Fernandez C, Jensen E, Flores J, et al. The immunophenotype of different immature, myeloid and B-cell lineage-committed CD34+ hematopoietic cells allows discrimination between normal/reactive and myelodysplastic syndrome precursors. Leukemia [Internet]. 2008;22(6):1175–83. Available from: http://dx.doi.org/10.1038/leu.2008.49
Comment 3. In Figure 7 and Figure 8, the author showed migration ability of thawed mononuclear cells from different donors. However, the author only used a rough FACS analysis pattern by FSC vs SSC to distinguish lympho/HSC- and mono-like cells. This data is not meaningful, simply because migration under VCAM1 /CXCL12 stimulation could not reflex the true effect of UBC cells in a living body. On the other hand, the proportion of granulocytes in samples, particularly neutrophils and eosinophils, can greatly affect the results. To show the ability of UCB HSPC cell migration, the author should provide more defined and detailed experiment results, such as co-culture with HUVECs or MSCs by trans-well method, and others.
Answer 3: To address this issue, as previously mentioned we included new experiments to test the clonogenic ability of the migrated cells and assess the functionality of thawed cells post-migration. This can be found in the new manuscript version on results section pages 15 and 16, lines from 474 to 496 and supplementary table S6.
In Figure 12A, we demonstrate that cells recovered in the lower chambers of the Transwell plates formed colonies across all donors tested. Notably, when normalizing CFU ability to the total events counted by flow cytometry, differences in clonogenic capacity were observed between donors, classified by their migratory profiles.
Your suggestion was invaluable, as it enabled us to highlight another functional distinction between the migratory profiles we have described. While we acknowledge that our classification of migratory behavior is somewhat rudimentary, in clinical settings, blocking the CXCR4/CXCL12 interaction is the primary method for mobilizing donor cells. Plerixafor, for example, is a mobilizing agent used in HPSC transplantation that inhibits CXCR4 and reversibly blocks its interaction with CXCL12, effectively mobilizing cells from bone marrow to peripheral blood. Similarly, in human and primate models, blocking the VCAM1/VLA4 axis yields comparable results.
Ruminski PG, Rettig MP, DiPersio JF. Development of VLA4 and CXCR4 antagonists for the mobilization of hematopoietic stem and progenitor cells. Biomolecules [Internet]. 2024;14(8):1003. Available from: http://dx.doi.org/10.3390/biom14081003
4, In conclusion part, the author should discuss more about establishing a criteria to distinguish the usable UCB samples from those un-usable ones.
Answer 4:
Based on your valuable suggestion, we have added the following paragraph to the conclusion of our paper on pages 21 and 22, lines from 717 to 731:
“Specifically, in the context of UCB transplantation, our results do not support the exclusion of any specific donor profile. While we observed that donors classified as Profile 1 exhibit a relatively higher proportion of HSPCs migrating toward CXCL12 and VCAM1, donor selection should also consider other critical factors, such as the viable TNC/kg dose, CD34+/kg count, and patient compatibility. However, these distinct migration patterns could serve as useful criteria for advanced therapies involving monocytes or lymphocytes derived from UCB cells. Selecting specific donor profiles for these therapies may enhance outcomes. We conclude that the inter-individual phenotypical and functional variability of banked UCB cells is extensive and highly dependent on the methodologies used for assessment. Thus, this variability should be explored and leveraged rationally based on the intended therapeutic application, including HSPC transplantation or advanced therapies derived from UCB cells.”
Reviewer 2 Report
Comments and Suggestions for Authors
[Major Issues]
Comment 1:
The reviewer appreciated the results of several flow cytometry experiments conducted under three conditions: freshly collected, post-volume reduction, and post-thawing. However, I have the following questions:
1. Figure 3 highlights a significant decrease in granulocytes under thawing conditions. Could this be attributed to a technical issue during the flow cytometry process, such as cell membrane damage caused by the thawing process, or does it indicate biological damage to the cells? To address this, please evaluate ROS generation to determine whether the cells are truly under stress.
2. In addition to changes in the number of cells, it would be important to assess any alterations in biological functions. Please evaluate cellular functions under thawing conditions, such as differentiation potential or the ability to induce inflammatory responses.
Comment 2:
1. The classification into migratory profiles 1 and 2 based on their responses to VCAM1 and CXCL12 is quite interesting. The criteria outlined in Figure 8C and the visualizations in Figures 8D and 8E were particularly well presented.
2. Since CXCR4 is the primary receptor for CXCL12 and is critical for regulating homing, could you verify the expression level of CXCR4 in each donor sample to confirm the classification of profiles 1 and 2?
3. Please provide additional explanation or references regarding the clinical implications of classifying into these two profiles, including predictions about treatment responses associated with each profile.
[Minor Issues]
Please explain the abbreviation “HLA” when it is first mentioned (in the introduction, second sentence).
On page 7 of 17, in the last paragraph, there appears to be confusion between the use of commas and periods when indicating the number of cells. For example, "400,000 cells" should be corrected.
Please adjust the scale of the y-axis in the graph in Figure 5 to a power of ten, consistent with another figure.
On page 10 of 17, in the last paragraph referring to Figures 8C and 8D, it seems the sentence should be revised to refer to Figures 8D and 8E. Kindly make this correction.
Author Response
Authors reply to reviewers:
Reviewer 2
Comment 1:
The reviewer appreciated the results of several flow cytometry experiments conducted under three conditions: freshly collected, post-volume reduction, and post-thawing. However, I have the following questions:
- Figure 3 highlights a significant decrease in granulocytes under thawing conditions. Could this be attributed to a technical issue during the flow cytometry process, such as cell membrane damage caused by the thawing process, or does it indicate biological damage to the cells? To address this, please evaluate ROS generation to determine whether the cells are truly under stress.
- In addition to changes in the number of cells, it would be important to assess any alterations in biological functions. Please evaluate cellular functions under thawing conditions, such as differentiation potential or the ability to induce inflammatory responses.
Answer to the Comment 1:
- We truly appreciate your suggestion of testing if cells were under stress, therefore, as an indirect response to the increase of ROS, we measured the early, late apoptosis and necrosis of cells. We described the percentage of cells isolated from UCB bags -instead of fragments of UCB samples- from four different donors which shows that some of the cells are in an apoptotic process. You will find this data in the new version of the manuscript: the materials and methods on page 5 (lines from 201 to 217) and the results from these assays on pages 15 and 16 (lines from 497 to 512) including Figure 9.
- Thank you for your kind suggestions to add more value to our paper. We could confirm that the thawing process preserved the hematopoietic progenitor and stem cell potential, even for VCAM1/CXCL12 transmigrated cells. To verify it, we added data from post-thawing clonogenicity assays. The methods were described on pages 4 and 5 (lines 181-199) while the results will be found on page 15 (lines from 474 to 496). Additional data was included in table S6 shown as an eClone value, and in figure 12 as CFU.
Comment 2:
- The classification into migratory profiles 1 and 2 based on their responses to VCAM1 and CXCL12 is quite interesting. The criteria outlined in Figure 8C and the visualizations in Figures 8D and 8E were particularly well presented.
- Since CXCR4 is the primary receptor for CXCL12 and is critical for regulating homing, could you verify the expression level of CXCR4 in each donor sample to confirm the classification of profiles 1 and 2?
- Please provide additional explanation or references regarding the clinical implications of classifying into these two profiles, including predictions about treatment responses associated with each profile.
Answer to the comment 2:
- We agree that your suggestion is a very important issue to address in our model, as CXCR4 expression could confirm the cellular classification of the donor within the Profile 1 or 2. However, the migratory pattern triggered by CXCL12 could be also modified by the expression of other receptors such as the CCR7, that also cross-regulate the sensitivity of the CXCR4 receptors. Furthermore, as in our experiments we are also using VCAM-1 as an adhesion molecule to stimulate the cell migration, the expression profile of the possible ligands would also help to show the mechanisms of this proposed classification. As the UCB samples that we have used in this study have been useful to classify the mentioned migratory profiles we are limiting its use to test all these features at once. We understand the importance of your suggestion, but it cannot be fulfilled within the 10 days deadline from the journal.
- To address the clinical implications of the described profiles, it is crucial to consider that blocking CXCR4 or VLA4 leads to the mobilization of HSPCs from the bone marrow to the peripheral blood. This phenomenon is detailed in a recent paper summarizing advancements in these pathways [39]. We anticipated that Profile 1, characterized by enhanced migratory capacity, would exhibit quicker engraftment. However, as our paper discusses and concludes, this trait should be integrated with other UCB cells characteristics. Building on your insightful suggestions, we have included our own speculations in the new version discussion (page 21, lines from 688 to 702) and conclusion sections (pages 21 and 22, lines from 717 to 731).
We carefully reviewed the entire manuscript and incorporated all the corrections you suggested, addressing each of your major and minor comments. We are deeply grateful for your invaluable feedback and thoughtful contributions, which have significantly enhanced the quality of our work.
Round 2
Reviewer 1 Report
Comments and Suggestions for Authors
Since the author provided reply that mostly answered my questions, I have no further comments.